Acute heat priming promotes short-term climate resilience of early life stages in a model sea anemone

http://orcid.org/0000-0002-2288-6389 Glass Benjamin H.
http://orcid.org/0009-0001-2290-7258 Jones Katelyn G.
http://orcid.org/0009-0003-6660-2848 Ye Angela C.
Dworetzky Anna G.
http://orcid.org/0000-0001-7371-4870 Barott Katie L. kbarott@sas.upenn.edu
Department of Biology, University of Pennsylvania , Philadelphia, Pennsylvania , United States of America
Sun Dong
Electronic publication date: 2023 Dec 5
Publication date: 2023
Volume: 11
Electronic Location ID: e16574
Received 2023 Sep 12; Accepted 2023 Nov 13
Copyright: © 2023 Glass et al.
Copyright year: 2023
Copyright holder: Glass et al.
License: This is an open access article distributed under the terms of the Creative Commons Attribution License, which permits unrestricted use, distribution, reproduction and adaptation in any medium and for any purpose provided that it is properly attributed. For attribution, the original author(s), title, publication source (PeerJ) and either DOI or URL of the article must be cited.
License URL: https://creativecommons.org/licenses/by/4.0/

Keywords: Nematostella vectensis, Hormetic priming, Environmental memory, Ocean warming, Climate change

Funding: National Institutes of Health (NIH) Predoctoral T32 HD083185 University of Pennsylvania American Fisheries Society Steven Berkeley Marine Conservation Fellowship National Science Foundation (NSF) 1923743 Foundation New Investigator Award KA2021-114797 This work was supported by the National Institutes of Health (NIH) Predoctoral T32 HD083185 to Benjamin H. Glass, funding from the University of Pennsylvania to Benjamin H. Glass, the American Fisheries Society Steven Berkeley Marine Conservation Fellowship to Benjamin H. Glass, the National Science Foundation (NSF) award 1923743 to Katie L. Barott, and the Charles E. Kaufman Foundation New Investigator Award KA2021-114797 to Katie L. Barott. The funders had no role in study design, data collection and analysis, decision to publish, or preparation of the manuscript.

==============================
Across diverse taxa, sublethal exposure to abiotic stressors early in life can lead to benefits such as increased stress tolerance upon repeat exposure. This phenomenon, known as hormetic priming, is largely unexplored in early life stages of marine invertebrates, which are increasingly threatened by anthropogenic climate change. To investigate this phenomenon, larvae of the sea anemone and model marine invertebrate Nematostella vectensis were exposed to control (18 °C) or elevated (24 °C, 30 °C, 35 °C, or 39 °C) temperatures for 1 h at 3 days post-fertilization (DPF), followed by return to control temperatures (18 °C). The animals were then assessed for growth, development, metabolic rates, and heat tolerance at 4, 7, and 11 DPF. Priming at intermediately elevated temperatures (24 °C, 30 °C, or 35 °C) augmented growth and development compared to controls or priming at 39 °C. Indeed, priming at 39 °C hampered developmental progression, with around 40% of larvae still in the planula stage at 11 DPF, in contrast to 0% for all other groups. Total protein content, a proxy for biomass, and respiration rates were not significantly affected by priming, suggesting metabolic resilience. Heat tolerance was quantified with acute heat stress exposures, and was significantly higher for animals primed at intermediate temperatures (24 °C, 30 °C, or 35 °C) compared to controls or those primed at 39 °C at all time points. To investigate a possible molecular mechanism for the observed changes in heat tolerance, the expression of heat shock protein 70 (HSP70) was quantified at 11 DPF. Expression of HSP70 significantly increased with increasing priming temperature, with the presence of a doublet band for larvae primed at 39 °C, suggesting persistent negative effects of priming on protein homeostasis. Interestingly, primed larvae in a second cohort cultured to 6 weeks post-fertilization continued to display hormetic growth responses, whereas benefits for heat tolerance were lost; in contrast, negative effects of short-term exposure to extreme heat stress (39 °C) persisted. These results demonstrate that some dose-dependent effects of priming waned over time while others persisted, resulting in heterogeneity in organismal performance across ontogeny following priming. Overall, these findings suggest that heat priming may augment the climate resilience of marine invertebrate early life stages via the modulation of key developmental and physiological phenotypes, while also affirming the need to limit further anthropogenic ocean warming.

Introduction

Anthropogenic ocean warming threatens the survival of marine invertebrates, and early life stages (e.g., larvae and juveniles) are particularly susceptible to heat stress (Ainsworth et al., 2016; Putnam et al., 2017; Pandori & Sorte, 2019; Byrne et al., 2020; Krämer, Iglesias-Prieto & Enríquez, 2022). Given the widespread negative effects experienced by marine invertebrates during and after marine heatwaves, in some cases including mass mortality (Dunn et al., 2004; Hughes et al., 2017; Goulet & Goulet, 2021; Pryor et al., 2021; Brown et al., 2022a), it is unclear whether these organisms possess the capacity to persist in the face of unprecedented, rapid climate change (Putnam et al., 2017; Torda et al., 2017; Byrne et al., 2020; Bairos-Novak et al., 2021). Within-generation acclimatization may promote resilience to ocean warming, and has gained increased attention in marine invertebrates including corals and sea anemones in recent years (Hawkins & Warner, 2017; Putnam et al., 2017; Putnam, 2021; Hackerott, Martell & Eirin-Lopez, 2021; Brown & Barott, 2022). While a unified terminology has yet to emerge for discourse surrounding within-generation acclimatization, modified responses to repeat exposure to abiotic stress are often termed “environmental memory” (Brown et al., 2015; Hackerott, Martell & Eirin-Lopez, 2021; Brown & Barott, 2022) or “legacy effects” (Wall et al., 2021; Wong et al., 2021), which can imply negative or positive responses to repeated stress. These phenomena may promote the persistence of marine invertebrates in a warming ocean, and are worthy of further investigation.

Beneficial acclimatization can develop via hormetic priming, in which exposure to low or intermediate doses of abiotic stress elicits physiological benefits, while higher doses lead to increasing net harm (Southam & Erlich, 1943; Costantini, Metcalfe & Monaghan, 2010). Hormesis gives rise to characteristic “U-” or “J-shaped” dose-response curves, in which a phenotype of interest (e.g., a physiological metric) peaks at intermediate values when quantified across a range of an abiotic stressor, giving the resulting curve the appearance of an inverted letter “U” or “J” (Southam & Erlich, 1943; Calabrese & Baldwin, 1999; Calabrese, 2004). For example, sublethal exposure to abiotic stressors (e.g., temperature or toxins) is known to have beneficial effects such as increased growth and improved stress tolerance upon repeat exposure in a broad diversity of taxa, from plants to animals (Costantini, Metcalfe & Monaghan, 2010; Berry & López-Martínez, 2020). In addition, hormetic priming in response to hyperthermal stress (i.e., heat or thermal priming) has been observed in plants (Migliore et al., 2010), insects (López-Martínez & Hahn, 2014), cnidarians (Martell, 2023), birds (Yahav & McMurtry, 2001), mammals (Caratero et al., 1998), and other taxa. In marine invertebrates, thermal priming has been best characterized in adult scleractinian corals, and typically involves a reduction in the bleaching response of symbiotic species to a subsequent exposure following an initial sublethal heat exposure (Hackerott, Martell & Eirin-Lopez, 2021; Brown & Barott, 2022). Beneficial priming in adult cnidarians has been linked to mechanisms including changes in gene expression (Rodriguez-Lanetty, Harii & Hoegh-Guldberg, 2009; Bay & Palumbi, 2015), cellular function (Majerova et al., 2021; McRae et al., 2021; Brown et al., 2022b), and metabolism (Gibbin et al., 2018). Despite growing knowledge surrounding heat priming in adult corals, we still lack an understanding of how priming effects vary across multiple priming temperatures (i.e., hormesis) and their role in other marine invertebrate taxa, particularly those that do not have algal endosymbionts. This gap in knowledge limits our understanding of how marine invertebrates and their particularly vulnerable early life stages may fare in future seas.

Current knowledge regarding the mechanisms and ecological implications of heat priming in marine invertebrates has largely arisen from studies on adult organisms, yet there is evidence that thermal priming also occurs during marine invertebrate early life stages (Byrne et al., 2020; Putnam, 2021; Hackerott, Martell & Eirin-Lopez, 2021). For example, heat priming of gametes from the reef-building corals Acropora cytherea and A. pulchra partially ameliorates heat-induced fertilization failure (Puisay et al., 2018, 2023). Additionally, in the coral Pocillopora damicornis, conditioning of brooded larvae to simulated ocean warming and acidification while still inside the adult colony increases settlement and survival following planulation (Putnam & Gates, 2015), as well as bleaching resistance and photosynthetic performance of algal endosymbionts under elevated temperatures (Jiang et al., 2023). However, in the corals A. tenuis and Montipora capitata, exposure to heat stress at the larval stage does not alter performance during a secondary heat exposure during the juvenile stage (Alexander et al., 2022; Hazraty-Kari et al., 2023), demonstrating that the effects of heat priming on early life stages are taxonomically variable and need to be characterized in a broader diversity of species and temperature regimes. As marine invertebrate early life stages are particularly sensitive to abiotic stressors (Przeslawski, Byrne & Mellin, 2015; Foo & Byrne, 2017; Byrne et al., 2020; Putnam, 2021), heat priming may be an important mechanism for the reproductive resilience of these taxa. However, key questions remain surrounding heat priming in early life stages. For example, it is necessary to further investigate what temperatures lead to beneficial priming as opposed to neutral or negative effects (i.e., hormesis), as most previous studies focus solely on comparisons between ambient and a single elevated (priming) temperature. Furthermore, molecular and cellular mechanisms underpinning temperature hormesis in marine invertebrate early life stages are unknown and in need of further investigation. These topics are of vital importance in a rapidly changing climate, particularly since spawning for many marine invertebrates occurs during the warmest months of the year (Harrison, 2011), meaning that larvae are being exposed to marine heatwaves with increasing frequency.

To address existing gaps in knowledge surrounding mechanisms of temperature hormesis in marine invertebrate early life stages, we used the widely studied sea anemone Nematostella vectensis as a model organism. N. vectensis inhabits temperate estuarine environments characterized by high diel and seasonal variability in abiotic conditions (Reitzel et al., 2013; Rosenau et al., 2021), and has evolved plastic mechanisms to support rapid acclimatization (Rivera et al., 2021; Baldassarre et al., 2022; Baldassarre, Reitzel & Fraune, 2023; Glass et al., 2023). For example, N. vectensis inhabits salt marshes along the Atlantic coast of New Jersey, USA, where temperatures can range from less than 0 °C to greater than 42 °C (annual median ~16–18 °C) and vary up to 20 °C within a single day, indicating that this species is a eurytherm (Reitzel et al., 2013; Rosenau et al., 2021). Additionally, while annual patterns of wild spawning in N. vectensis are currently uncharacterized, gametogenesis requires only ~2 weeks in this species and has been observed at culture temperatures between 16–24 °C (Stefanik, Friedman & Finnerty, 2013; Rivera et al., 2021), suggesting that N. vectensis may be able to spawn throughout the year in the wild. Given that N. vectensis displays wide temperature tolerance (Reitzel et al., 2013), plasticity in thermal tolerance in response to both abiotic and biotic environmental factors (Reitzel et al., 2013; Rivera et al., 2021; Baldassarre et al., 2022), and can easily be induced to spawn (Hand & Uhlinger, 1992; Fritzenwanker & Technau, 2002; Stefanik, Friedman & Finnerty, 2013), this species is an important and tractable model in which to investigate environmental memory and hormesis in early life stages (Darling et al., 2005; Layden, Rentzsch & Röttinger, 2016; Martindale, 2022). Additionally, given its wide geospatial distribution and similar stress responses to other invertebrate taxa, including ctenophores and echinoderms (Przeslawski, Byrne & Mellin, 2015; Foo & Byrne, 2017; Byrne et al., 2020; Martindale, 2022; Glass et al., 2023), N. vectensis can provide insight into the capacity for environmental memory in the early life stages of marine invertebrates more broadly. Finally, N. vectensis is a common model species for the study of early development, so characterizing priming responses in this species will provide important knowledge regarding the effects of environmental factors on development, while also laying the foundation for future work on this topic. Here, larvae from N. vectensis were exposed to control (18 °C) or elevated (24 °C, 30 °C, 35 °C, or 39 °C) temperatures for 1 h at 3 days post-fertilization (DPF). Larvae were then returned to culture at 18 °C and assessed for growth, development, and various physiological metrics before, during, and after settlement (4, 7, and 11 DPF, respectively). To investigate a possible mechanism of modified heat tolerance following priming, the expression of heat shock protein 70 (HSP70) in primed larvae was quantified at 11 DPF. Additionally, to gain insight into the temporal extent of priming effects, a second larval cohort was also primed and followed through 6 weeks post-fertilization. Overall, this study constitutes a novel examination of heat priming in a model sea anemone, and suggests that dose-dependent priming may augment marine invertebrate resilience to anthropogenic ocean warming via hormesis by the modulation of developmental and physiological phenotypes during early life stages.

Materials and Methods

Adult collection, culture, and sexual reproduction

Adult Nematostella vectensis sea anemones were collected from a salt marsh in Brigantine, New Jersey in the fall of 2020. Following transport to the laboratory, anemones were kept in 12 parts per thousand (ppt) artificial seawater (1/3-ASW; Spectrum Brands, Madison, WI, USA) at 18 °C in a dark incubator (Boekel Scientific, Feasterville Trevose, PA, USA). Animals were fed twice per week with Artemia nauplii (Brine Shrimp Direct, Ogden, UT, USA) with water changes occurring every 2 weeks for approximately 2.5 years. Spawning was induced using a standard method for N. vectensis (Hand & Uhlinger, 1992; Fritzenwanker & Technau, 2002; Stefanik, Friedman & Finnerty, 2013), which entailed exposing anemones (N = 200 adults across four containers) to light and elevated temperatures (24 °C) for 14 h followed by transfer to room temperature (∼18–19 °C), after which spawning occurred within 1–2 h. As culture containers housed both male and female anemones, eggs were left to fertilize during spawning and then transferred to a plastic dish with ~25 mL new 1/3-ASW. Fertilized embryos were then held at 18 °C in the dark for 3 days.

Priming treatment and larval culture

At 3 days post-fertilization (DPF), 200 swimming planula larvae were pipetted into each of 15 conical tubes (15 mL), for a total of three replicate tubes of 200 larvae (N = 600 larvae priming temperature−1) for each of the five priming temperature treatments: 18 °C (control), 24 °C, 30 °C, 35 °C, and 39 °C (Fig. 1). At a duration of 1 h, these temperatures are sublethal for N. vectensis larvae (Rivera et al., 2021; Glass et al., 2023), and representative of the range of temperatures and diel variability that can occur in this species’ natural habitat (Reitzel et al., 2013). Tubes were placed in water baths (Thermo Fisher Scientific, Waltham, MA, USA) set to each treatment temperature for 1 h. Following the treatment period, tubes containing larvae were poured into petri dishes (15 mL capacity) and a partial water change was performed by aspirating and replacing ~7.5 mL (~50%) 1/3-ASW; dishes were then held at 18 °C in the dark for the remainder of the experiment. The day after the priming treatment, one dish of larvae primed at 39 °C displayed 100% mortality along with dense overgrowth of an unidentified microbe, and was therefore removed from the experiment. All other larvae were left unfed and in the same water for the remainder of the short-term experiment (11 days in total). For the long-term experiment, this entire procedure was repeated for a second cohort of larvae, which were held at 18 °C for 6 weeks following priming at 3 DPF. In both experiments, control animals were kept at 18 °C for the entire experiment duration, having gone through the priming treatment but without a change in temperature.

Figure 1 Experimental design and representative images.

The figure depicts the experimental design with representative images of larvae and juveniles over time. At 0 days post-fertilization (DPF), adult Nematostella vectensis sea anemones (N = 200 adults across four containers) were induced to spawn and their gametes left to fertilize. Developing embryos were kept at 18 °C. At 3 DPF, larvae were divided into 15 groups of 200, and each group was primed at either 18 °C (control), 24 °C, 30 °C, 35 °C, or 39 °C for 1 h (N = 3 groups per priming temperature), then returned to 18 °C for the remainder of the experiment. At 4, 7, and 11 DPF, animals in each group were assayed for growth, respiration, total protein, and heat tolerance. The fertilization and priming process was repeated for a second cohort of larvae; images were collected weekly through 6 weeks post-fertilization (WPF), and heat tolerance was assayed at 6 WPF.

Image collection and quantification of growth and development

Each day following the priming treatment (4–11 DPF), dishes containing larvae were individually removed from the incubator and photographed in a single, haphazardly chosen region under a dissecting microscope (Leica MZ12; Leica, Wetzlar, Germany) with a camera attachment (Retiga R3 CCD), after which they were returned to the incubator (Fig. 1). Each photograph contained at least 20 larvae, and microscope settings were unchanged between dishes; a ruler was also photographed each day using the same microscope settings. To quantify larval growth, images were analyzed in FIJI (Schindelin et al., 2012); the ruler image was used to set the scale, and then the line tool was used to individually measure the lengths along the longest axis of at least 20 larvae per photograph (N = 40–60 larvae priming temperature−1 time point−1; Fig. S1). Next, developmental progression was quantified from the images by counting the number of larvae clearly past the planula stage (i.e., not a homogenous oval shape), which was divided by the total number of larvae in each image and then converted to a percentage of “post-planula” larvae (N = 17–129 larvae priming temperature−1 time point−1).

Respiration and total protein measurements

At 4, 7, and 11 DPF, 15 larvae per culture dish (N = 30–45 larvae priming temperature−1 time point−1) were transferred to a 24-well plate (N = 1 well dish−1) with oxygen sensor spots (Loligo Systems, Viborg, Denmark). All wells were filled to capacity (80 μL) with 1/3-ASW, and 3–4 wells were also filled with culture water without larvae to serve as controls. The plate was sealed with an adhesive plate cover and placed on a calibrated PreSens SensorDish Reader (Precision Sensing, Regensburg, Germany) at room temperature (~21 °C) under ambient lighting. The oxygen concentration (μmol O2 L−1) in each well was recorded every 15 s for 1 h, during which conditions remained normoxic (>180 μmol O2 L−1). Following respiration measurements, larvae in each well (N = 15 larvae) were transferred to 1.5 mL tubes, seawater was removed, and the tubes were frozen and stored at −80 °C. Tubes containing larvae were later thawed on ice, and 60 μL of 1x tris-NaCl-EDTA lysis buffer supplemented with dithiothreitol, protease inhibitor cocktail (Thermo Fisher Scientific, Waltham, MA, USA), and phosphatase inhibitors (Roche, Basel, Switzerland), were added to each tube. Next, larvae were lysed in a water bath sonicator (Diagenode UCD-200; Diagenode Inc., Denville, NJ, USA) at 4 °C for 5 min with a 30:60 s on:off cycle. Following sonication, tubes were centrifuged (1,500 × g for 5 min at 4 °C) and the protein concentrations of the supernatants were determined in triplicate using a Bradford assay with a bovine serum albumin standard curve. Due to defects in the plastic of the 1.5 mL tubes used for total protein assays, six of the 42 protein samples were lost during processing. However, at least one sample was processed for each time point and priming temperature combination, with the exception of 24 °C at 7 DPF. To determine respiration rates, the rate of oxygen consumption for each well was determined as the slope of a linear best-fit line of the oxygen level in the well over time, and the average rate for the control wells was subtracted from the wells containing larvae. The absolute values of the slopes were then converted to pmol O2 minute−1 μg protein−1 by multiplying by 1,000 and the volume of the wells (80 μL), and dividing by the total protein.

Heat tolerance measurements

Larval heat tolerance was determined at 4, 7, and 11 DPF using previously established methods (Rivera et al., 2021; Glass et al., 2023). Specifically, six larvae from each culture dish were exposed to each of the following peak temperatures: 39 °C, 40 °C, 41 °C, 42 °C, or 43 °C, yielding 2–3 dose-response curves (N = 30 larvae curve−1) per priming temperature. Specifically, individual larvae were pipetted into wells of a 96-well PCR plate with 100 μL of 1/3-ASW, and plates were sealed with an adhesive plate cover to prevent evaporation during the assay. Thermocyclers (Thermo Fisher Scientific, Waltham, MA, USA) were used to generate the heat ramps, and were programmed as follows: (i) 1 min at 25 °C; (ii) 4 min at 30 °C; (iii) 4 min at 38 °C; (iv) 1 h at peak temperature (39–43 °C); (v) 4 min at 38 °C; (vi) 4 min at 30 °C; and (vii) infinite hold at 22 °C. Following completion of the heat ramp, the plates were uncovered and placed at 18 °C for 48 h, after which larvae were scored as dead (visible tissue lysis) or alive (no visual abnormalities; often swimming). For each dose-response curve, the proportion of larvae surviving at each peak temperature was calculated by dividing the number of larvae surviving by the total number of larvae treated. Data pertaining to proportion survival were used to calculate lethal temperature 50s (LT50s) for each dose-response curve as detailed below.

Western blotting for heat shock protein 70 (HSP70)

Following the extraction of proteins and quantification via a Bradford assay (see above), equal amounts (μg) of protein in lysis buffer from each group of larvae collected at 11 DPF (N = 2–3 groups of 15 larvae priming temperature−1) were combined with Laemlli buffer (Bio-Rad, Hercules, CA, USA), denatured at 70 °C for 15 min, and loaded at a target amount of 2.02 μg protein well−1 into a 4–12% tris-glycine gel. Next, electrophoresis was performed for 30 min at 60 V followed by 1 h at 120 V, and proteins were then transferred to a polyvinylidene fluoride membrane (100 V for 100 min at 4 °C). Following transfer, the membrane was blocked for 1 h in blocking buffer (5% w/v bovine serum albumin in tris-buffered saline (TBS) with 0.1% v/v Tween-20 (TBST)) and incubated with 0.55 μg mL−1 polyclonal antibodies for HSP70 (Novus Biologicals, Centennial, CO, USA), which were chosen based on the target epitope’s ability to bind to all five known isoforms of N. vectensis HSP70 (Waller et al., 2018; Knighton et al., 2019). Next, the membrane was washed (3 × 10 min with TBST) and a secondary antibody (anti-rabbit IgG with horseradish peroxidase) was added for 1 h before final washing (3 × 10 min with TBST followed by 1 × 10 min with TBS), treatment with chemiluminescence reagents (Thermo Fisher Scientific, Waltham, MA, USA), and imaging on an Amersham Imager 600 (General Electric, Boston, MA, USA). After initial imaging, the membrane was probed for β-tubulin using 2.5 μg mL−1 monoclonal antibodies (Cell Signaling Technology, Danvers, MA, USA) and reimaged.

Long-term growth and heat tolerance experiments

In order to characterize the persistence of heat priming effects on growth and heat tolerance, a second cohort of larvae was produced 2 weeks after the initial spawning by the same adult population and exposed to the same priming treatment at 3 DPF as described above. Following priming, larvae were kept in culture through 6 weeks post-fertilization (WPF) at 18 °C in the dark, where they progressed to the juvenile stage and began to grow tentacles (Fig. 1). Animals were fed twice per week (i.e., every 3–4 days) after settlement (~7 DPF) with homogenized Artemia nauplii. Specifically, 3 mL of live Artemia nauplii were homogenized in a 15 mL conical using a rotostator at 20,000 rpm for 10 s, and 200 μL of the resulting slurry was added to each culture dish. Within 2–3 h after feeding, a partial water change was performed for each dish by aspirating and replacing 5–7 mL (33–50%) of 1/3-ASW. Images were collected weekly as described above to characterize long-term effects of priming on growth, and the number of tentacles possessed by each juvenile in the images was also quantified. After 6 weeks, juvenile heat tolerance was determined via a single-temperature heat challenge at 42 °C due to fewer juveniles in the long-term priming experiment vs the short-term experiment. Specifically, eight juveniles from each culture dish (N = 24 juveniles priming temperature−1) were exposed to a heat ramp peaking at 42 °C for 1 h then returned to 18 °C. While juveniles were intended to be monitored at intervals over 48 h to generate dose-response curves over time, >60% mortality for all groups at 21 h precluded this procedure, so the assay was concluded at that time. Finally, the number of surviving juveniles was divided by the number treated to obtain the proportion surviving.

Data analysis

All data were analyzed using R version 4.2.1 (R Core Team, 2022) in RStudio (RStudio Team, 2020). First, proportion survival data from larval heat tolerance assays were used to create dose-response curves represented by two-parameter log-logistic functions using the package drc (Ritz et al., 2015), and an LT50 was determined for each curve using the package chemCal (Ranke, 2022). Data pertaining to growth (body column length and juvenile tentacle number), development (post-planula larvae), respiration rates, and LT50s were analyzed using linear mixed-effect models with priming temperature, DPF/WPF, and their interaction as effects and culture dish as a random effect. For total protein, the interaction between priming temperature and DPF was not significant, so a linear model relating protein to the additive combination of priming temperature and DPF was used for this metric. Data pertaining to normalized HSP70 expression at 11 DPF were also analyzed using a linear model with priming temperature as a fixed effect. Additionally, to investigate the correlation between HSP70 expression and LT50 at 11 DPF, a linear model was built with HSP70 expression as a fixed effect. For juvenile survival following heat shock at 6 WPF, a linear model was created with priming temperature as a fixed effect. All models were confirmed to meet relevant assumptions (linearity, statistical independence of errors, homoscedasticity of errors, and normality of the error distribution) via visual inspection of diagnostic plots (residuals vs fitted values, residuals vs leverage, scale-location, and normal Q-Q, respectively) generated using the “plot()” function. Models were then subjected to Type II (protein, HSP70, and juvenile survival following heat shock) or Type III (all others) ANOVAs to test for the significance of fixed effects, followed by Tukey’s Honest Significant Difference (HSD) post-hoc tests to test for the significance of pairwise comparisons where appropriate. Additional packages used for analysis include: ggplot2 (Wickham, 2016), lme4 (Bates et al., 2015), car (Fox & Weisberg, 2019), plyr (Wickham, 2011), dplyr (Wickham et al., 2023), emmeans (Lenth et al., 2023), and ggpubr (Kassambara, 2023).

Results

Short-term growth and metabolism following thermal priming

Body column length was significantly influenced by the interaction between DPF and priming temperature (p < 0.001; Figs. 1 and 2). At 4 DPF, lengths were statistically indistinguishable for larvae across all priming temperatures (Table 1). At 7 DPF, larvae were significantly larger compared to 4 DPF for all priming temperatures except 39 °C (p < 0.05). Additionally, body lengths at 7 DPF were significantly longer for larvae primed at intermediately elevated temperatures (24 °C, 30 °C, and 35 °C) compared controls (18 °C) and those primed at 39 °C (p < 0.05). At 11 DPF, larvae were again significantly larger than at 7 DPF for all priming temperatures except 39 °C (p < 0.05). Furthermore, body lengths at 11 DPF were again significantly greater for larvae primed at intermediately elevated temperatures compared to controls and those primed at 39 °C (p < 0.05). Larval biomass, as estimated from total protein (expressed as μg larva−1 ± SEM; Table 1) was not significantly influenced by DPF (p = 0.094) or priming temperature (p = 0.465; Fig. 3A). Respiration rate (expressed in pmol O2 minute−1 μg protein−1; Table 1) was also not significantly influenced by DPF (p = 0.107), priming temperature (p = 0.57), or their interaction (p = 0.809; Fig. 3B).

Figure 2 Effects of heat priming on short-term larval growth by day.

Body column length (in mm) of Nematostella vectensis larvae (N = 40–60 larvae priming temperature−1 day−1) over priming temperatures (°C) and days post-fertilization (DPF). DPF and the interaction between DPF and priming temperature were significant (p < 0.05) linear model terms. Points with error bars represent means with standard error, and are colored by DPF. Data from days for which physiological data were also collected are represented by line types consistent with other plots (4 DPF = solid, 7 DPF = long dash, and 11 DPF = dot-dash), while data for all other days are connected by dotted lines.

Table 1 Means with standard error for all metrics (short-term experiment).

Days post-fertilization	Priming temperature (°C)	Body column length (mm)	Post-planula larvae (%)	Total protein (μg larva−1)	Respiration rate (pmol O2 min−1 μg protein−1)	LT50 (°C)	HSP70 (relative intensity)	
4	18	0.23 ± 0.01	0 ± 2.3	0.262 ± 0.051	6.45 ± 1.87	41.98 ± 0.1	NA	
4	24	0.23 ± 0.01	0 ± 2.3	0.269 ± 0.078	9.13 ± 2.83	42.24 ± 0.04	NA	
4	30	0.23 ± 0.01	0 ± 2.3	0.3 ± 0.054	6.95 ± 1.99	42.19 ± 0.04	NA	
4	35	0.23 ± 0.01	0 ± 2.3	0.365 ± 0.024	6.33 ± 0.23	42 ± 0.08	NA	
4	39	0.24 ± 0.01	0 ± 2.3	0.434 ± NA	5.6 ± NA	41.33 ± 0.08	NA	
7	18	0.26 ± 0.01	40.4 ± 2.3	0.313 ± 0.09	6.78 ± 1.12	40.14 ± 0	NA	
7	24	0.28 ± 0.01	63.2 ± 2.3	NA	NA	40.34 ± 0.08	NA	
7	30	0.29 ± 0.01	61.1 ± 2.3	0.363 ± 0.064	4.78 ± 1.35	40.42 ± 0.08	NA	
7	35	0.29 ± 0.01	59.2 ± 2.3	0.330 ± 0.018	4.8 ± 0.47	40.27 ± 0	NA	
7	39	0.23 ± 0.01	13.6 ± 2.3	0.463 ± 0.023	4.16 ± 0.38	40.14 ± 0	NA	
11	18	0.38 ± 0.01	100 ± 2.3	0.406 ± 0.041	5.69 ± 0.6	41.44 ± 0.12	0.165 ± 0.008	
11	24	0.4 ± 0.01	100 ± 2.3	0.465 ± NA	3.41 ± NA	41.92 ± 0	0.165 ± 0.008	
11	30	0.41 ± 0.01	100 ± 2.3	0.388 ± 0.045	4.32 ± 0.41	41.84 ± 0.08	0.189 ± 0.006	
11	35	0.4 ± 0.01	100 ± 2.3	0.416 ± 0.041	4.89 ± 0.33	41.52 ± 0.14	0.2 ± 0.011	
11	39	0.26 ± 0.01	57.9 ± 2.3	0.399 ± 0.14	4.98 ± 1.45	41.25 ± 0	0.299 ± 0.033	

Figure 3 Effects of heat priming on larval metabolism and development.

(A) Total protein (expressed as μg larva−1; N = 30–45 larvae priming temperature−1 time point−1) and (B) respiration (in pmol O2 minute−1 μg protein−1; N = 30–45 larvae priming temperature−1 time point−1) of Nematostella vectensis larvae over priming temperature (°C) and DPF (line types). (C) The percentage of larvae progressed past the planula stage (i.e., post-planula larvae; N = 40–60 larvae priming temperature−1 day−1) over priming temperature (°C) and days post-fertilization (DPF). DPF and the interaction between DPF and priming temperature were significant (p < 0.05) linear model terms. The legend applies to all plots, and the points with error bars represent means with standard error.

Developmental progression following thermal priming

The percentage of larvae that had developed past the planula stage was significantly influenced by the interaction between DPF and priming temperature (p < 0.001; Fig. 3C). No larvae had developed past the planula stage at 4 DPF; however, all groups contained larvae that had progressed beyond the planula stage by 7 DPF (Fig. 1, Table 1). At this time, significantly higher percentages of larvae had developed past the planula stage for groups primed at intermediately elevated temperatures (24 °C, 30 °C, and 35 °C) compared to controls (18 °C) or those primed at 39 °C (p < 0.05). Furthermore, control groups (18 °C) showed significantly higher percentages of post-planula development than those primed at 39 °C (p < 0.05) at 7 DPF. By 11 DPF, 100% of larvae had developed past the planula stage for controls and groups primed at intermediately elevated temperatures, whereas only 57.9 ± 2.3% of larvae primed at 39 °C had progressed beyond the planula stage by that time.

Larval heat tolerance following thermal priming

Larval heat tolerance (expressed at lethal temperature 50 (LT50) in °C) was derived from each dose-response curve (Fig. 4A). For all assays, 100% of larvae survived 1 h at the lowest treatment temperature (39 °C), and survival decreased with increasing treatment temperature such that 0% of larvae survived after treatment at 43 °C (Fig. 4A). LT50 was significantly influenced by the interaction between DPF and priming temperature (p < 0.001; Fig. 4B). At 4 DPF, LT50s were significantly higher for controls and larvae primed at intermediately elevated temperatures (24 °C, 30 °C, and 35 °C) compared to those primed at 39 °C (p < 0.05; Table 1). LT50s were significantly lower at 7 DPF compared to 4 DPF across all priming temperatures (p < 0.05). Additionally, LT50s at 7 DPF were significantly higher for larvae primed at intermediately elevated temperatures compared to controls and those primed at 39 °C (p < 0.05). LT50s at 11 DPF were significantly higher compared to 7 DPF (p < 0.05) and significantly lower compared to 4 DPF (p < 0.05) across all priming temperatures. Additionally, LT50s at 11 DPF were significantly higher for larvae primed at 24 °C and 30 °C compared to all other priming temperatures (p < 0.05).

Figure 4 Effects of heat priming on larval heat tolerance.

(A) Survival (%) of Nematostella vectensis larvae assayed for heat tolerance via exposure to short heat ramps peaking between 39–43 °C at 4, 7, and 11 days post-fertilization (N = 2–3 groups of 30 larvae priming temperature−1 time point−1). The points with lines are colored by priming temperature and correspond to raw data for each subgroup; not all points and lines are visible where data overlap. (B) Heat tolerance (N = 2–3 groups of 30 larvae treatment−1 time point−1), expressed as lethal temperature 50 (LT50) in °C, over priming temperature (°C) and DPF (line type). DPF, priming temperature, and the interaction between DPF and priming temperature were significant (p < 0.05) linear model terms. The points with error bars represent means with standard error.

Expression of heat shock protein 70 (HSP70) following thermal priming.

At 11 DPF, controls (18 °C) and larvae primed at intermediately elevated temperatures (24 °C, 30 °C, and 35 °C) expressed a single isoform of HSP70 (Fig. 5A). Larvae primed at 39 °C (only two culture dishes represented due to mortality in the third dish, see above) also expressed the same isoform, as well as an additional faint band at a slightly higher molecular weight (Fig. 5A). When quantified and normalized to β-tubulin, HSP70 protein expression was significantly influenced by priming temperature (p < 0.001; Fig. 5B), with levels increasing as priming temperature increased (Table 1). Finally, HSP70 expression displayed a significant, negative correlation with LT50 (p = 0.03; Fig. 5C).

Figure 5 Effects of heat priming on HSP70 expression and relationship to heat tolerance.

(A) Western blot for heat shock protein 70 (HSP70) at 11 days post-fertilization (DPF) in Nematostella vectensis larvae primed at different temperatures (above lanes; N = 2–3 groups of 15 larvae priming temperature−1). The magnified portion shows the bands for larvae primed at 39 °C more closely, enabling visualization of doublet bands; an image of β-tubulin (loading control) is also displayed. (B) HSP70 expression (arbitrary units) normalized to β-tubulin over priming temperature in °C; priming temperature was a significant (p < 0.05) linear model term. The points with error bars represent means with standard errors. (C) Correlation between lethal temperature 50 (LT50) and priming temperature, both in °C, with a best-fit line of data and shaded 95% confidence interval. The points represent the unprocessed data, and the inset displays the p value for an analysis of variance (ANOVA) testing the significance of the correlation.

Long-term growth, development, and heat tolerance

In the long-term experiment, body column length was significantly influenced by the interaction between weeks post-fertilization (WPF) and priming temperature (p < 0.001; Fig. 6A). Lengths generally increased over time from 1–6 WPF (Table 2), with the exception of animals primed at 39 °C, which ceased to grow after 2 WPF. Indeed, the only significant difference in lengths for animals primed at 39 °C after 1 WPF was between 3 and 6 WPF, with juveniles at 6 WPF displaying significantly smaller lengths than those at 3 WPF. At 1, 3, 4, 5, and 6 WPF, animals primed at intermediately elevated temperatures (24 °C, 30 °C, and 35 °C) displayed significantly larger body column lengths compared to controls (18 °C), which, along with the intermediately elevated temperatures, displayed significantly larger lengths than animals primed at 39 °C. Tentacle number was significantly influenced by the interaction between WPF and priming temperature (p < 0.001; Fig. 6B). As with length, tentacle number generally significantly increased across 1–6 WPF (Table 2). At 1, 3, and 4 WPF, tentacle number was relatively consistent for controls (18 °C) and animals primed at temperatures between 24–35 °C, which all had significantly more tentacles than animals primed at 39 °C for the same weeks (p < 0.05). Additionally, at 5 and 6 WPF, animals primed at 30 °C and 35 °C had more tentacles compared to controls and those primed at all other temperatures, while animals primed at 24 °C had significantly more tentacles than controls and those primed at 39 °C at 6 but not 5 WPF (p < 0.05). Juvenile heat tolerance at 6 WPF was significantly influenced by priming temperature (p = 0.007; Fig. 6C). Specifically, all juveniles displayed 100% survival at 6 h post-exposure (Table 2), but by 21 h post-exposure, controls and juveniles primed 24 °C displayed significantly higher survival rates than those primed at 35 °C or 39 °C (p < 0.05).

Figure 6 Long-term effects of heat priming on growth and heat tolerance.

(A) Body column length (in mm) of Nematostella vectensis larvae and juveniles (N = 12–60 animals priming temperature−1 week−1) over priming temperatures (°C) and weeks post-fertilization (WPF). WPF, priming temperature, and their interaction were significant (p < 0.05) linear model terms. (B) Tentacle number of larvae and juveniles (N = 12–60 animals priming temperature−1 week−1) over priming temperatures (°C) and WPF. WPF, priming temperature, and their interaction were significant (p < 0.05) linear model terms. (C) Juvenile heat tolerance at 6 WPF expressed as the percent of animals surviving 21 h after a 1-h heat shock at 42 °C (N = 3 groups of eight animals priming temperature−1). Priming temperature was a significant (p < 0.05) linear model term. On all plots, the points with error bars represent means with standard error, and are colored by WPF.

Table 2 Means with standard error for all metrics (long-term experiment).

Weeks post-fertilization	Priming temperature (°C)	Body column length (mm)	Tentacle number	Juvenile survival after heat shock (%)	
1	18	0.41 ± 0.01	0.8 ± 0.1	NA	
1	24	0.45 ± 0.01	0.7 ± 0.1	NA	
1	30	0.43 ± 0.02	0.4 ± 0.1	NA	
1	35	0.47 ± 0.02	0.5 ± 0.1	NA	
1	39	0.3 ± 0.01	0.1 ± 0.1	NA	
2	18	0.65 ± 0.02	1.6 ± 0.2	NA	
2	24	0.68 ± 0.02	1 ± 0.1	NA	
2	30	0.6 ± 0.01	1.2 ± 0.2	NA	
2	35	0.69 ± 0.02	1.6 ± 0.2	NA	
2	39	0.55 ± 0.02	0.6 ± 0.2	NA	
3	18	0.72 ± 0.02	3.2 ± 0.1	NA	
3	24	0.8 ± 0.03	3.4 ± 0.1	N	
3	30	0.82 ± 0.03	3.4 ± 0.1	NA	
3	35	0.9 ± 0.03	3.3 ± 0.1	NA	
3	39	0.64 ± 0.02	1.3 ± 0.2	NA	
4	18	0.91 ± 0.03	3.2 ± 0.1	NA	
4	24	0.95 ± 0.03	3 ± 0.2	NA	
4	30	0.99 ± 0.04	3.1 ± 0.2	NA	
4	35	0.95 ± 0.03	2.9 ± 0.2	NA	
4	39	0.56 ± 0.02	0.9 ± 0.2	NA	
5	18	0.82 ± 0.02	3.4 ± 0.2	NA	
5	24	0.89 ± 0.03	3.4 ± 0.2	NA	
5	30	0.98 ± 0.04	4 ± 0.2	NA	
5	35	0.97 ± 0.03	4.2 ± 0.2	NA	
5	39	0.55 ± 0.02	1.3 ± 0.2	NA	
6	18	0.97 ± 0.05	4.4 ± 0.1	29.2 ± 4.2	
6	24	1.2 ± 0.04	4.7 ± 0.2	33.3 ± 4.2	
6	30	1.25 ± 0.03	5.1 ± 0.2	16.7 ± 8.3	
6	35	1.19 ± 0.06	4.5 ± 0.2	4.2 ± 4.2	
6	39	0.5 ± 0.03	1.8 ± 0.3	4.2 ± 4.2	

Means with standard error and pairwise comparisons

Means with standard error for all metrics are listed in Tables 1 (short-term experiment) and 2 (long-term experiment). Statistical testing information for pairwise comparisons (degrees of freedom, T ratios, and p values) is available in Supplemental Files.

Discussion

Here, we found that sublethal heat stress led to hormesis during early life stages in the model sea anemone Nematostella vectensis. Growth, development, and heat tolerance all displayed hormetic responses, being augmented in larvae primed at intermediately (24 °C, 30 °C, and 35 °C) but not extremely (39 °C) elevated temperatures compared to controls (18 °C). Additionally, expression of heat shock protein 70 (HSP70) was significantly higher in larvae primed at elevated temperatures even 8 days after priming, yet was negatively correlated with heat tolerance, suggesting that HSP70 protein expression alone cannot explain the modulation of larval hyperthermal performance following priming. Furthermore, the effects of priming on growth and heat tolerance persisted beyond the larval stage, up to at least 6 weeks post-fertilization (WPF). These findings represent a novel contribution to a growing body of knowledge surrounding hormesis in marine invertebrates, and suggest that dose-dependent hormetic priming can have long-term, beneficial effects on marine invertebrate early life stages, which may augment the resilience of these imperiled organisms to anthropogenic ocean warming.

Heat priming affected growth and metabolism

Priming at intermediately elevated temperatures (24 °C, 30 °C, and 35 °C) led to increased body column lengths in animals at 7 and 11 days post-fertilization (DPF). This response may be beneficial for fitness, as body size is positively correlated with survival and reproductive output in a broad diversity of organisms (Brown, Marquet & Taper, 1993). However, it is also possible that increased growth after priming at intermediately elevated temperatures might incur a trade off with other traits. For example, the use of metabolic resources for increased growth could result in decreases in important homeostatic mechanisms, such as morphological regulation (Negri, Marshall & Heyward, 2007; Randall & Szmant, 2009). However, we did not observe tradeoffs between growth and any of the other traits measured in this study, and suggest this topic for future study. The observation of a parabolic relationship between temperature and growth is common for biological rate processes (Schulte, Healy & Fangue, 2011), and is congruent with previous findings that larval growth in N. vectensis larvae peaks following sustained exposure to intermediately elevated temperatures (Reitzel et al., 2013). However, it was surprising that hyperthermal priming for just 1 h at 3 DPF was sufficient to produce augmented growth, as it was previously believed that only sustained exposure to elevated temperatures would produce this response (Reitzel et al., 2013). Additionally, it was intriguing that hormetic effects of heat priming on short-term growth were largely absent in fed larvae at 1 and 2 WPF. Regardless of priming temperature, fed larvae displayed body column lengths similar to those of starved larvae primed at intermediately elevated temperatures (24 °C, 30 °C, and 35 °C) at 7 and 11 DPF. This trend suggests that, in fed larvae, no additional growth benefit was gained following heat priming, possibly because nutritional resources had already allowed these larvae to grow to a stage-specific size optimum. Strikingly, despite the absence of hormetic responses in growth at 1 and 2 WPF in fed larvae, juvenile body column lengths and tentacle numbers (i.e., developmental rate) later displayed classical hormetic U-curves—being increased at intermediately elevated priming temperatures—that persisted through 6 WPF. These findings demonstrate that a single hour of exposure to hyperthermal stress at 3 DPF can have long-term consequences for growth and development, which may be dampened early in ontogeny when external nutritional resources are abundant. It was also striking that larvae primed at 39 °C displayed no growth after 2 WPF, particularly given that N. vectensis can transiently experience temperatures in the range of 39–43 °C and diel variability in the range of 20 °C day−1 in its natural estuarine habitats (Reitzel et al., 2013). This result suggests that N. vectensis larvae are likely to be threatened by anthropogenic global warming, which is increasing the frequency and severity of acute temperature spikes in global estuaries (Najjar et al., 2000; Najjar, Patterson & Graham, 2009; Oczkowski et al., 2015; Shi & Hu, 2022). On the whole, results reported here are in contrast with previous work in other species; for example, larvae of the broadcast spawning coral Acropora tenuis exposed to elevated temperatures for 14 days showed no differences in size from naive larvae (Hazraty-Kari et al., 2023). These findings raise the possibility that the temperatures chosen for priming may have been outside of the optimal range or duration to induce a beneficial response in A. tenuis, that nutritional resources from heterotrophic feeding or algal endosymbionts may have dampened priming responses, or that priming effects on growth vary by species. Future research should investigate comprehensive phenotypic profiles following heat priming at better resolved temperature and temporal scales in both fed and unfed animals to further characterize the effects of elevated temperatures on early life stages.

Total protein content in N. vectensis larvae was unaffected by priming across days and priming treatments. Protein is a major component of organismal biomass, and higher total protein content generally indicates increased amounts of cellular material in a given tissue (Giese, 1966). The stability of protein content was particularly surprising given the observed changes in body column length, as increased growth might be expected to correlate with increased protein content per larva (Frieder et al., 2018). However, the total protein content of tissues in marine invertebrates is not simply a function of size, but rather varies nonlinearly with cellular metabolism (Giese, 1966; Frieder et al., 2018). Indeed, no differences in respiration rates were observed among larvae primed at different temperatures, suggesting a possible uncoupling of metabolism from the observed changes in growth and development. Changes in respiration rates following heat priming have rarely been investigated in marine invertebrate larvae, though elevated temperatures can lead to decreased respiration rates in larvae of the stony corals Pocillopora damicornis (Edmunds, Cumbo & Fan, 2011; Putnam et al., 2013), P. acuta (Kitchen et al., 2022), Porites astreoides (Olsen et al., 2013), Seriatopora hystrix (Edmunds, Cumbo & Fan, 2011), and Stylophora pistillata (Edmunds, Cumbo & Fan, 2011), whereas larvae of the corals Montipora capitata and Lobactis scutaria display no change in respiration rates under short-term heat stress (Kitchen et al., 2022). The lack of differences in protein and respiration observed here demonstrate that N. vectensis larvae displayed metabolic resilience to transient heat stress, likely reflecting evolved adaptations to its abiotically variable and sometimes extreme native habitats (Reitzel et al., 2013; Rivera et al., 2021; Glass et al., 2023). These results also suggest that effects of heat priming on growth and development observed here resulted from mechanisms possibly unrelated to metabolism, which should be further investigated. For example, it is possible that priming modified the epigenome (e.g., DNA methylation or histone modifications), as has been observed in corals (Ishida-Castañeda, Iguchi & Sakai, 2023), ascidians (Hawes et al., 2018), and oysters (Wang et al., 2021) following heat stress, resulting in differential gene expression that precipitated changes in growth and development but not metabolism. Alternatively, it is possible that changes in metabolic rates did occur in response to priming but on shorter timescales than investigated here, such that a return to fixed, ambient temperatures masked these responses. In other words, N. vectensis may use short-term plasticity in metabolic rates as a mechanism to rapidly acclimatize to its highly variable environment. Future studies could investigate the impacts of temperature fluctuation following priming at finer temporal scales to further characterize the role of metabolism in rapid acclimatization. Overall, these results contribute novel insights into the interplay between growth, development, and metabolism following transient exposure to heat stress in marine invertebrate larvae, and suggest that future studies could take advantage of the wealth of resources available for N. vectensis to further explore molecular mechanisms underpinning these phenomena.

Heat priming affected development, heat tolerance, and HSP70 expression

Priming of larvae at intermediately elevated (24 °C, 30 °C, or 35 °C), but not high (39 °C), temperatures accelerated the development of N. vectensis. Faster progression past the planula stage might improve the fitness of N. vectensis larvae and help meet increased metabolic needs under elevated temperatures, as this would shorten the period between fertilization and the independent acquisition of nutrients via feeding upon the development of the mouth during settlement. Indeed, faster progression past the planula stage corresponded with increased tentacle numbers at 6 WPF for juveniles primed at intermediately elevated temperatures. This may have resulted from increased feeding capacity due to faster developmental rates in these animals, given that tentacle development is feeding-dependent in N. vectensis (Ikmi et al., 2020). Interestingly, accelerated development past the planula stage did not directly result in increased tentacle numbers at 1 and 2 WPF in fed larvae. It is possible that the ability to obtain a small amount of food even prior to the development of tentacles helped meet increased metabolic demands following priming, such that tentacle growth per se was not an energetic priority for these larvae as it was for starved animals. These results demonstrate that, in the context of heat priming, there is not always a linear relationship between development and food availability in N. vectensis larvae, complicating previous findings from adults of this species. In ectotherms, increased temperatures typically accelerate developmental progression due to elevated activity of cellular enzymes (Pineda et al., 2012; Stoks, Geerts & De Meester, 2014; Šargač et al., 2022). While it is known that development is influenced by temperature in N. vectensis larvae (Reitzel et al., 2013), these results are particularly surprising in that a single hour at elevated temperatures at 3 DPF was sufficient to accelerate early development. Additionally, it is intriguing that nearly half of the larvae primed at 39 °C did not progress past the planula stage by 11 DPF, as this further corroborates a classical hormetic response (Costantini, Metcalfe & Monaghan, 2010; Berry & López-Martínez, 2020). Given that N. vectensis occasionally experiences temperatures in the range of 39–43 °C in its natural habitat (Reitzel et al., 2013), it is possible that the delay in development following heat shock is an adaptive mechanism to prolong the pelagic planula stage when conditions are nonoptimal, possibly allowing larvae to disperse to a more suitable environment for settlement. Severe temperature stress has also been shown to slow or halt development in early stages of larval development in the coral Pseudodiploria strigosa (Bassim & Sammarco, 2003), whereas larvae of the corals Acropora palmata and Astroides calycularis display accelerated development under elevated temperatures as observed here, but with a concurrent increase in developmental abnormalities (Randall & Szmant, 2009; Carbonne et al., 2022). These conflicting results underscore that the dose and duration of heat stress, in addition to external nutrient availability, likely interact to determine developmental outcomes. Furthermore, thermal priming of Acropora pulchra oocytes for 1 h prior to fertilization accelerates larval development at ambient temperatures (Puisay et al., 2018), suggesting that effects of priming on development occur in both gametes and larvae. Future research investigating the molecular mechanisms of accelerated development in marine invertebrates and any downstream fitness implications are warranted, particularly given the increasing threat of anthropogenic ocean warming.

As with development, the heat tolerance of N. vectensis larvae in this study displayed a hormetic response to priming temperature. In N. vectensis, thermal tolerance is plastic and influenced by developmental stage (Reitzel et al., 2013), microbiome composition (Baldassarre et al., 2022; Baldassarre, Reitzel & Fraune, 2023), and parental (particularly maternal) temperature history (Rivera et al., 2021). Developmental differences in heat tolerance across ontogeny observed here recapitulate previous findings from this species (Reitzel et al., 2013), with larvae displaying highest heat tolerance at 4 DPF, which then declined at 7 DPF, followed by an increase at 11 DPF. This is possibly because maternally provided nutritional resources augment heat tolerance in the earliest days following fertilization, followed by rapid consumption of these resources and a concomitant decrease in heat tolerance, which then increases again when metabolic demands begin to subside. Based on this study, hyperthermal priming is an additional mechanism influencing heat tolerance in N. vectensis larvae, with a single hour of exposure to intermediately elevated temperatures at 3 DPF being sufficient to increase larval LT50s by up to ~0.8 °C, approximating differences in LT50s observed between natural populations of this species along its latitudinal cline (Reitzel et al., 2013; Rivera et al., 2021; Baldassarre et al., 2022). As with the observed responses in growth and development, it is possible that heat-induced increases in thermal tolerance result from mechanisms evolved by N. vectensis to adapt to high diel and annual variability in its natural estuarine habitats. However, the hormetic nature of this response suggests that this mechanism may be inadequate to protect N. vectensis larvae under intensifying estuary warming (Najjar, Patterson & Graham, 2009). Additionally, hormetic benefits of hyperthermal priming on heat tolerance were lost in the long term, with juveniles assessed at 6 WPF displaying decreases in thermal tolerance with increasing priming temperature. In addition to further demonstrating that heat tolerance is heterogeneous during development, these findings suggest that certain benefits of heat priming wane across ontogeny. Future research should investigate the molecular mechanisms underpinning heterogeneity in the effects of heat priming on thermal tolerance over time, in addition to characterizing whether heat priming also impacts tolerance under more naturalistic, fluctuating temperature regimes. In other hexacorals, heat priming at early life stages can also improve thermal tolerance. For example, priming of gametes from the reef corals Acropora cytherea and A. pulchra partially ameliorates fertilization failure under increased temperatures (Puisay et al., 2018, 2023). As with the other metrics investigated here, the effects of priming on heat tolerance may vary by species and depend on physiological and ecological factors such as endosymbiosis with algae (Baird, Guest & Willis, 2009; Schnitzler et al., 2012; Cumbo, van Oppen & Baird, 2018; Ng, Chui & Ang, 2019; Kitchen et al., 2022).

The expression of HSP70 proteins may have played a role in priming-induced changes in heat tolerance. HSP70 protein expression at 11 DPF significantly increased with increasing priming temperature, building upon previous findings that HSP70 transcript abundance is increased immediately after heat shock in larvae of N. vectensis and the corals Acropora millepora, A. palmata, and A. pruinosa (Rodriguez-Lanetty, Harii & Hoegh-Guldberg, 2009; Polato, Altman & Baums, 2013; Rivera et al., 2021; Chui et al., 2023). Given that HSP70 mRNA in hydra has a half-life on the order of hours following acute heat shock (Brennecke, Gellner & Bosch, 1998), whereas HSP70 proteins generally show high stability in cells (Theodorakis & Morimoto, 1987), it is possible that HSP70 proteins were translated in abundance during and shortly after the priming treatment, and then persisted for several days. In Drosophila, HSP70 protein levels increase following heat shock, but decline along with gains in thermal tolerance within hours after return to ambient temperatures (Danxi & Duncan, 2008). Given these findings, more work is needed to better understand HSP expression dynamics in N. vectensis. Intriguingly, larvae primed at 39 °C displayed a doublet band on the Western blot for HSP70, which has been previously observed for N. vectensis HSP70 immediately following heat shock (Waller et al., 2018; Knighton et al., 2019) and indicates interaction with misfolded client proteins to prevent cytotoxic aggregation. The persistent upregulation of HSP70 in N. vectensis following heat priming could impair larval growth and fitness, as production of HSP70 proteins directs cellular resources away from translation (Heckathorn et al., 1996a, 1996b; Rodriguez-Lanetty, Harii & Hoegh-Guldberg, 2009), which would be particularly detrimental in a developing organism. Indeed, the redirection of cellular resources towards the heat shock response may help explain the developmental delays and decreases in thermal tolerance observed in larvae primed at 39 °C. Further supporting this hypothesis, increased HSP70 levels at 11 DPF in animals primed at 35 °C and 39 °C were correlated with decreases in thermal tolerance at 6 WPF for animals primed at the same temperatures, suggesting that lasting cellular damage and/or energetic deficits were incurred as a result of persistent HSP70 upregulation. Similarly, overexpression of HSP70 in fruit fly larvae and HSP22 in adult fruit flies has deleterious effects, including decreases in stress tolerance and lifespan (Krebs & Feder, 1997; Bhole, Allikian & Tower, 2004), and HSP (over)expression is associated with the depression of protein synthesis and cell proliferation (DiDomenico, Bugaisky & Lindquist, 1982; Sanchez et al., 1992; Feder & Hofmann, 1999), demonstrating that HSPs can have negative effects in certain cellular contexts. In combination, these results affirm that heat tolerance is an emergent phenotype likely resulting from interactions between genetic, epigenetic, physiological, and other factors, which can change across development. Future research should investigate the longitudinal effects of acute heat priming on organismal fitness (e.g., reproductive output) to characterize the temporal limits of priming’s costs and benefits, and to further determine whether this mechanism may serve as a pathway to resilience for marine invertebrates under ocean warming.

Conclusions

Overall, our findings show that heat priming can have significant effects on growth, development, and physiology across marine invertebrate early life stages. Future research should more fully characterize phenotypic profiles following larval heat priming to define the viability of this phenomenon and its underlying mechanisms for promoting resilience in warming seas. Finally, given the pervasive and persistent negative effects of exposure to even transient high temperature stress, our study maintains the importance of rapid interventions for limiting the extent of future ocean warming.

Supplemental Information

Supplemental Information 1 Methods for determination of body column length.

Cropped images of Nematostella vectensis larvae and juveniles at various lengths and stages of development, and a cropped image of a ruler used to determine the body column lengths. Images were collected using identical microscope settings, then the ruler image was used to set scale and a line was drawn along the major axis of the body column of each animal (black lines with arrowheads) to determine the length. White text displays the values resulting from the example measurements.

Click here for additional data file.

Supplemental Information 2 Body column length pairwise comparisons.

Click here for additional data file.

Supplemental Information 3 Post-planula larvae pairwise comparisons.

Click here for additional data file.

Supplemental Information 4 LT50 pairwise comparisons.

Click here for additional data file.

Supplemental Information 5 HSP70 pairwise comparisons.

Click here for additional data file.

Supplemental Information 6 Long-term length pairwise comparisons.

Click here for additional data file.

Supplemental Information 7 Tentacle number pairwise comparisons.

Click here for additional data file.

Supplemental Information 8 Long-term heat tolerance pairwise comparisons.

Click here for additional data file.

Supplemental Information 9 HSP70 and tubulin uncropped blot.

Click here for additional data file.

The authors would like to acknowledge Dr. Kristen T. Brown for providing guidance in analyzing dose-response curve data and constructive feedback on figures. The authors would also like to thank two anonymous reviewers for constructive feedback that helped improve the manuscript.

Additional Information and Declarations

Competing Interests

Author Contributions

Data Availability

The authors declare that they have no competing interests.

Benjamin H. Glass conceived and designed the experiments, performed the experiments, analyzed the data, prepared figures and/or tables, authored or reviewed drafts of the article, and approved the final draft.

Katelyn G. Jones performed the experiments, authored or reviewed drafts of the article, and approved the final draft.

Angela C. Ye performed the experiments, authored or reviewed drafts of the article, and approved the final draft.

Anna G. Dworetzky performed the experiments, authored or reviewed drafts of the article, and approved the final draft.

Katie L. Barott conceived and designed the experiments, performed the experiments, analyzed the data, prepared figures and/or tables, authored or reviewed drafts of the article, and approved the final draft.

The following information was supplied regarding data availability:

All original data and code are available on Dryad: Glass, Benjamin et al. (2023). Data and code for: Acute heat priming promotes short-term climate resilience of early life stages in a model sea anemone [Dataset]. Dryad. https://doi.org/10.5061/dryad.f4qrfj724.

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
