# Peer review of "Acute heat priming promotes short-term climate resilience of early life stages in a model sea anemone"

_PeerJ, doi:10.7717/peerj.16574_

## Round 0.1 · original submission · Major Revisions

Both reviewers gave excellent reviews, and the second reviewer in particular gave good technical advice. Please read them and revise your manuscript.
Look forward to seeing the revised draft soon.

Reviewer 1 ·

Basic reporting

Summary:
The authors investigated heat priming in a sea anemone Nematostella vectensis. An introduction of beneficial acclimatization via hormesis is well described for related taxa, preceding explanation of a literature gap – heat priming in hexacoral larvae remains understudied as well as the cellular-molecular underpinnings of hormesis in this species. To this end, the authors deployed two experiments in which anemone larvae remained under ambient conditions (18°C) or were primed to four temperatures (24°C, 30°C, 35°C, or 39°C) for 1 hour at 3 days post-fertilization preceding a return to 18°C for (1) short-term phenotypic tracking until 11 days post-fertilization and (2) long term phenotypic tracking until 6 weeks post-fertilization. Under the short-term experiment, unfed larvae were photographed each day to measure lengths and developmental progression. Respiration rates, total protein, and heat tolerance were measured at days 4, 7, and 11 days post fertilization – the short-term experiment also concluded with quantification of HSP70 at 11 days post-fertilization. Under the long-term challenge, fed larvae were reared for 6 weeks until heat tolerance was measured. Heat tolerance in both experiments was defined as the LT50 following a 1-hour exposure to peak temperature in groups from each priming treatment – in the short term experiment larvae were exposed to five peak temperatures (39°C, 40°C, 41°C, 42°C, or 43°C) whereas larvae were exposed to a single peak temperature in the long-term term challenge (42°C). Authors found unfed larvae were larger and developed faster (post-planula stage) on the order of days to 4-11 days post-priming to intermediately elevated temperatures (24°C, 30°C, 35°C). Heat tolerance exhibited a similar pattern during the short-term experiment (intermediate heat primed = higher LT50), however this trend did not persist to six weeks post-fertilization. Lastly, a higher LT50 at 11 days-post fertilization was attributed with greater expression of HSP70. Overall, the authors executed an impressive design to achieve J-curves indicative of hormesis. The figures represent the data very well and will strengthen readers’ text-to-visual understanding of authors’ rationale and findings. Considering the rise of ‘priming’-related study in non-models, I suspect high impact spanning taxa and field. A few areas for expansion are encouraged, including some background, term consistency, and discrepancies between methods deployed in the two experiments. I suggest that moderate revisions and no further experiments are required.

Experimental design

review #s 1 and 2 in 'Validity of the findings' section

Validity of the findings

Major Comments
Numbered list suggesting changes that require substantial work/time than a minor edit – in no particular order of importance

(1) A few points for expansion of the discussion (A-B):

A. Major difference between the short- and long-term experiments require expanded text in the discussion and/or methods for clarity and explanation of physiological response.

o How does food affect priming:

Repeat datasets in the presence and absence of food can be interpreted from the short-term experiment (unfed) alongside the initial weeks of the long-term experiment (fed). Comparisons between these data suggest that the initial dose response curve on order of days to two-weeks post-fertilization was dependent concurrent starvation.

Figure 1 – body column length for starved*priming at 7 and 11 days
- hormetic response curve

Figure 6 – body column length for fed*priming at 7 and 14 days
- no hormetic response curve

Food supply shows a clear role here and should be addressed – especially considering the emphasis of the short-term experiment findings in this study. I suggest expansion of the discussion to include the importance of food/energetic limitations on priming. One consideration is an possible expedited J-curve under starvation (11 DPF) that was otherwise absent under food supply (2 WPF) and arose at later life stages (5-6 WPF).

Lines 429-433: Great discussion point here and a possible location for adding text. Why does priming elicited faster progression to planula stage under starvation (to 11 DPF) but not under food supply (to 2 WPF/14 DPF)?

B. Heat tolerance - clarity/rationale of methods and suggestion for expanding discussion.

o Clarity/rationale for methodology changes:

In the short-term challenge, mortality was assessed in sea anemone larvae 48-hours post exposure (at 4, 7, and 11 DPF). In the long-term challenge, mortality was assessed at 6 and 21 hours post exposure and only one temperature (at 6 weeks post fertilization). What was the rationale for this change?

o Suggestion for expanded discussion, dependence on duration post-priming:

Comparison between short- and long-term trial suggest that heat tolerance difference between life stages and/or duration post priming event.

Lines 456-471: Section speaks to the importance of the higher heat tolerance days following priming, a striking result very well depicted in Figure 4. This pattern represented by those prime under intermediately elevated temperatures did not persist to 6 weeks (Figure 6). I suggest expansion here on the duration/transience of heat tolerance or life-stage specificity of this pattern.

(2) Major strengths of this study are the well-executed design to produce dose-dependent response curves indicative of hormesis. The authors do well to acknowledge background pertaining to environmental acclimatization in related taxa. To strengthen this paper and broaden its reach, I suggest adding a brief and concise general definition of hormesis citing foundational literature and the expected U/J-shaped dose-dependent curves in response to environmental stimuli.

(3) HSP70 relationship with heat tolerance

Lines 495-502: Authors have a hypothesized direction for molecular-phenotypic effects of priming – those with higher LT50 (hyperthermal performance) should have higher HSP70, however results suggest otherwise.

Added literature review and discussion is requested here. Though high protein chaperone activity can ameliorate homeostasis during challenges, overactivity can represent an energetic expenditure precursory to cell death.

Additional comments

Minor Comments
Numbered list with reference to Line numbers of small changes. For example, grammatical changes, citation issues, word choice, clarity of sentence construction/meaning, etc.

Lines 25: define how heat tolerance was determined in brief

Lines 31: unclear without reading methods that the 6-week larvae are not the same individuals/trial. Consider revising to state the two experiments.

Line 216: amounts – concentration? volume?

Line 218: 2.02 ug protein per well – is this a targeted number or the resulting average calculated downstream? If the later, provide standard error.

Consistency in reference to N.
The term ‘subgroup’ is introduced in Line 143 and given at around 5 cases to supplement description of N for data collection. In this manner the reader needs to calculate the true N -- alternatively the N per priming treatment is easily understood in Line 247: N = 8 juveniles from each culture dish (N=24 juveniles priming temperature-1). Consider revising throughout.

Lines 252-273: Please add all tests for model assumptions where relevant.

Lines 271-273: Greatly appreciate your time and effort to acknowledge R package authors!

Line 302: This is the first case when an effect is shown for larvae primed to 24°C, 30°C, and 35°C – later this is described as larvae primed to ‘intermediately elevated temperature’. I suggest defining this term here and use it consistently throughout to describe findings with this same pattern.

Line 332: WPF already introduced in Line 237

Lines 276-357 (Results section): In all cases, significant main effects are provided when interaction terms are also significant. Though not a strong advocate for omitting all explanation of main effects in presence of an interaction, however I suggest considering omission when main effects provide redundancy to what is evident from the posthoc interaction terms. One example being in Lines 341-343 in which tentacle number was affected by time, priming and time*priming interaction – the main priming effect is not explained but I assume the posthoc shows all priming temperatures > 39C. This result is highlighted by the interaction term with the added benefit of WPF resolution.

Line 424: What are these some relevant examples?

Line 480: What does this interaction mean in context of a heat shock response?

Reviewer 2 ·

Basic reporting

This is a well-written elegant paper describing heat priming on larval development in the sea anemone Nematostella vectensis. All aspects of the work are solid. The topic is extremely interesting and impactful for a number of reasons. The results have implications for how marine animals will react to ever increasing temperatures. It provides insight into the development and physiology of one of the most important invertebrate research models in many fields. It also provides some insight into the effect of heat stress more broadly into non-anemone hexacorals including reef corals. Below I list a few ways the paper could be improved.

The abstract and discussion stress the importance of this study for better understanding hexacoral biology, with a nod towards coral reefs. Given the importance of Nematostella as a model animal, and the limited information about how temperature affects Nematostella development, the implications of the results on Nematostella itself is important. The paper as written reads a little like studying Nematostella is only worthwhile if it says something about corals. It would be strange to do these experiments in a hard coral and discuss mostly about the implications on sea anemones.

It would help readability if the 18-degree primed Nematostella were referred to as a control throughout. It could also help to mention explicitly that these Nematostella were essentially kept at 18-degrees (ambient) for the entire study, but that they had gone through the priming process minus the temperature change. If not read carefully, it seems like the 18-degree is a treatment rather than a control and that there are in fact no controls.

Line 257: "body column length"
>> to maximize the reproducibility of this study, please include a figure showing anemones of various lengths with a ruler, the length determined from the sample measurement, and arrows pointing to the boundaries of the body column.

Line 277: "The day after the priming treatment, one dish of larvae primed at 39°C displayed 100% mortality along with dense overgrowth of an unidentified microbe, and was thus removed from the experiment."
>> Starting off the results with this is problematic for several reasons. One, is that it took me a few paragraphs later to realize that this dish did not represent all larvae primed at 39 and caused me confusion. It's also odd to start off the results section with an outlier. Even though it is a result, I think it might go better in the methods section or in the supplement, with a small reference to it later in the results.

>> A discussion of what might be happening with the heat shock proteins would be helpful. For example, given what is known about the stability and degradation of HSP70 protein, could it be that these proteins were translated during the priming and have stuck around? Or, given what is known about the stability and degradation of HSP70 mRNA, could it be that the mRNA is stabilized and perhaps that this mRNA is being translated continuously post-priming? Could residual HSP70 having a direct effect on the results in this study?

>> More background is needed on native habitats and Nematostella spawning habits. What are the median temperatures of tide pools in New Jersey salt marshes? This could be extrapolated from reported ocean temperatures. Is there information on whether Nematostella spawn in winter months? Is there information on the temperature ranges in which Nematostella will spawn in laboratory settings? Introducing and discussing these data in the context of these findings would be helpful. If the data are not available, it would be helpful at least to mention how connecting the results to the natural biology of the animal would be a natural future direction.

>> Fixed temperatures are important for studies like these. However, Nematostella experiences high fluctuations of temperature in their natural environment. How might these results be interpreted in light of these two aspects?

Line 370: dose-dependent hormetic priming can have long-term, beneficial effects for marine invertebrate early life stages, which may augment the resilience of these imperiled organisms to anthropogenic ocean warming.
>> Given that 24-degrees may be closer to median temperatures of the tide pools in New Jersey Salt Marshes during peak summer months than 18-degrees, and that it is possible that Nematostella only spawn in their natural environment during these months (unless there is data suggesting otherwise--see previous point), might it be that priming is closer to reality for these animals and what is being observed in the 18-degree priming is more of a treatment than a control? The paper would be stronger if the results were put into an ecological context by discussing the data in relation to temperature regimens in which Nematostella are usually exposed.

>> These results suggest a number of future directions. It would be great to explore a bit of the possibilities in the discussion.

Experimental design

The experimental design is solid.

Validity of the findings

The findings are valid.

---

## Round 0.2 · accepted · Accept

I have carefully examined the authors' responses to both review comments, and I do not believe that a second round of review is necessary.